# Transductive Linear Probing: A Novel Framework for Few-Shot Node Classification

**Zhen Tan**[*]
Arizona State University
ztan36@asu.edu

**Song Wang**[*]
University of Virginia
sw3wv@virginia.edu

**Kaize Ding**[*]
Arizona State University
kding9@asu.edu

**Jundong Li**
University of Virginia
jundong@virginia.edu

**Huan Liu**
Arizona State University
huanliu@asu.edu

## Abstract

Few-shot node classification is tasked to provide accurate predictions for nodes from novel classes with only few representative labeled nodes. This problem has drawn tremendous attention for its projection to prevailing real-world applications, such as product categorization for newly added commodity categories on an E-commerce platform with scarce records or diagnoses for rare diseases on a patient similarity graph. To tackle such challenging label scarcity issues in the non-Euclidean graph domain, meta-learning has become a successful and predominant paradigm. More recently, inspired by the development of graph self-supervised learning, transferring pretrained node embeddings for few-shot node classification could be a promising alternative to meta-learning but remains unexposed. In this work, we empirically demonstrate the potential of an alternative framework, *Transductive Linear Probing*, that transfers pretrained node embeddings, which are learned from graph contrastive learning methods. We further extend the setting of few-shot node classification from standard fully supervised to a more realistic self-supervised setting, where meta-learning methods cannot be easily deployed due to the shortage of supervision from training classes. Surprisingly, even without any ground-truth labels, transductive linear probing with self-supervised graph contrastive pretraining can outperform the state-of-the-art fully supervised meta-learning based methods under the same protocol. We hope this work can shed new light on few-shot node classification problems and foster future research on learning from scarcely labeled instances on graphs.

## 1 Introduction

Graph Neural Networks (GNNs) [1–4] are a family of neural network models designed for graph-structured data. In this work, we concentrate on GNNs for the node classification task, where GNNs recurrently aggregate neighborhoods to simultaneously preserve graph structure information and learn node representations. However, most GNN models focus on the (semi-)supervised learning setting, assuming access to abundant labels [5, 6]. This assumption could be practically infeasible due to the high cost of data collection and labeling, especially for large graphs. Moreover, recent works have manifested that directly training GNNs with limited nodes can result in severe performance degradation [7–9]. Such a challenge has led to a proliferation of studies [6, 10–12] that try to learn fast-adaptable GNNs with extremely scarce known labels, *i.e., Few-Shot Node Classification* (FSNC) tasks. Particularly, in FSNC, there exist two disjoint label spaces: *base classes* are assumed to contain substantial labeled nodes while target *novel classes* only contain few available labeled nodes. If the target FSNC task contains $N$ novel classes with $K$ labeled nodes in each class, the problem is

---

[*]Equal contribution.

Z. Tan et al., Transductive Linear Probing: A Novel Framework for Few-Shot Node Classification. *Proceedings of the First Learning on Graphs Conference (LoG 2022)*, PMLR 198, Virtual Event, December 9–12, 2022.

denoted as an $N$-way $K$-shot node classification task. Here the $K$ labeled nodes are termed as a *support set*, and the unlabeled nodes are termed as a *query set* for evaluation.

Currently, *meta-learning* has become a prevailing and successful paradigm to tackle such a shortage of labels on graphs. Inspired by the way humans learn unseen classes with few samples via utilizing previously learned prior knowledge, a typical meta-learning based framework will randomly sample a number of *episodes*, or *meta-tasks*, to emulate the target $N$-way $K$-shot setting [7]. Based on this principle, various models [7–13] have been proposed, which makes meta-learning a plausible default choice for FSNC tasks. On the other hand, despite the remarkable breakthroughs that have been made, meta-learning based methods still have several limitations. **First**, relying on different arbitrarily sampled meta-tasks to extract transferable meta-knowledge, meta-learning based frameworks suffer from the piecemeal knowledge issue [14]. That being said, a small portion of the nodes and classes are selected per episode for training, which leads to an undesired loss of generalizability of the learned GNNs regarding nodes from unseen novel classes. **Second**, the feasibility for sampling meta-tasks is based on the assumption that there exist sufficient base classes where substantial labeled nodes are accessible. However, this assumption can be easily overturned for real-world graphs where the number of base classes can be limited, or the labels of nodes in base classes can be inaccessible. In a nutshell, these two concerns motivate us to design an alternative framework for meta-learning to cover more realistic scenarios.

Inspired by [15, 16], we postulate that the key to solving FSNC is to learn a generalizable GNN encoder. We validate this postulation by a motivating example in Section 2.3. Then, without the episodic emulation, the proposed novel framework, *Transductive Linear Probing* (TLP), directly transfers pretrained node embeddings for nodes in *novel classes* learned from *Graph Contrastive Learning* (GCL) methods [17–23], and fine-tunes a separate linear classifier with the support set to predict labels for unlabeled nodes. GCL methods are proven to learn generalizable node embeddings by maximizing the representation consistency under different augmented views [17, 18, 23, 24]. If the representations of nodes in novel classes are discriminative enough, probing them with a simple linear classifier should provide decent accuracy. Based on this intuition, we propose two instantiations of the TLP framework in this paper: TLP with the self-supervised form of GCL methods and TLP with the supervised GCL counterparts. We evaluate TLP by transferring node embeddings from various GCL methods to the linear classifier and compare TLP with meta-learning based methods under the same evaluation protocol. Moreover, we examine the effect of supervision during GCL pretraining for target FSNC tasks to further analyze what role labels from base classes play in TLP.

Throughout this paper, we aim to shed new light on the few-shot node classification problem through the lens of empirical evaluations of both the "old" meta-learning paradigm and the "new" transductive linear probing framework. The summary of our contributions is as follows:

**New Framework** We are the first to break with convention and precedent to propose a new framework, transductive linear probing, as a competitive alternative to meta-learning for FSNC tasks.

**Comprehensive Study** We perform comprehensive reviews on current literature and the research community and conduct a large-scale study on six widely-used real-world datasets that cover different scenarios in FSNC: (1) a sufficient number of base classes with substantial labeled nodes in each class, (2) a sufficient number of base classes with no labeled nodes in each class, (3) a limited number of base classes with substantial labeled nodes in each class, and (4) a limited number of base classes with no labeled nodes in each class. We evaluate all the compared methods under the same protocol.

**Findings** We demonstrate that despite the recent advances in few-shot node classification, meta-learning based methods struggle to outperform TLP methods. Moreover, the TLP-based methods with self-supervised GCL can outperform their supervised counterparts and those meta-learning based methods even if all the labels from base classes are inaccessible. This signifies that without label information, self-supervised GCL can focus more on node-level structural information, which results in better node representations. However, TLP also inherits its limitation for scalability due to the large memory consumption of GCL, which makes it hard to deploy on extremely large graphs. Based on those observations, we identify that improving adaptability and scalability are the promising directions for meta-learning based and TLP-based methods, respectively.

Our implementations for experiments are released[2]. We hope to facilitate the sharing of insights and accelerate the progress on the goal of learning from scarcely labeled instances on graphs.

---

[2] https://github.com/Zhen-Tan-dmml/TLP-FSNC.git

## 2 Preliminaries

### 2.1 Problem Statement

Formally, given an attributed network $\mathcal{G} = (\mathcal{V}, \mathcal{E}, \mathbf{X}) = (\mathbf{A}, \mathbf{X})$, where $\mathcal{V}$ denotes the set of nodes $\{v_1, v_2, ..., v_n\}$, $\mathcal{E}$ denotes the set of edges $\{e_1, e_2, ..., e_m\}$, $\mathbf{X} = [\mathbf{x}_1; \mathbf{x}_2; ...; \mathbf{x}_n] \in \mathbb{R}^{n \times d}$ denotes all the node features, and $\mathbf{A} = \{0, 1\}^{n \times n}$ is the adjacency matrix representing the network structure. Specifically, $\mathbf{A}_{j,k} = 1$ indicates that there is an edge between node $v_j$ and node $v_k$; otherwise, $\mathbf{A}_{j,k} = 0$. The few-shot node classification problem assumes that there exist a series of target node classification tasks, $\mathcal{T} = \{\mathcal{T}_i\}_{i=1}^{I}$, where $\mathcal{T}_i$ denotes the given dataset of a task, and $I$ denotes the number of such tasks. We term the classes of nodes available during training as base classes (i.e., $\mathbb{C}_{base}$) and the classes of nodes during target test phase as novel classes (i.e., $\mathbb{C}_{novel}$) and $\mathbb{C}_{base} \cap \mathbb{C}_{novel} = \varnothing$. Notably, under different settings, labels of nodes for training (i.e., $\mathbb{C}_{base}$) may or may not be available during training. Conventionally, there are few labeled nodes for novel classes $\mathbb{C}_{novel}$ during the test phase. The problem of few-shot node classification is defined as follows:

**Definition 1.** *Few-shot Node Classification: Given an attributed graph $\mathcal{G} = (\mathbf{A}, \mathbf{X})$ with a divided node label space $\mathbb{C} = \{\mathbb{C}_{base}, \mathbb{C}_{novel}\}$, we only have few-shot labeled nodes (support set $\mathbb{S}$) for $\mathbb{C}_{novel}$. The task $\mathcal{T}$ is to predict the labels for unlabeled nodes (query set $\mathbb{Q}$) from $\mathbb{C}_{novel}$. If the support set in each target (test) task has $N$ novel classes with $K$ labeled nodes, then we term this task an $N$-way $K$-shot node classification task.*

The goal of few-shot node classification is to learn an encoder that can transfer the topological and semantic knowledge learned from substantial data in base classes ($\mathbb{C}_{base}$) and generate discriminative embeddings for nodes from novel classes ($\mathbb{C}_{novel}$) with limited labeled nodes.

### 2.2 Episodic Meta-learning for Few-shot Node Classification.

Episodic meta-learning is a proven effective paradigm for few-shot learning tasks [25–32]. The main idea is to train the neural networks in a way that emulates the evaluation conditions. This is hypothesized to be beneficial for the prediction performance on test tasks [25–27, 33]. Based on this philosophy, many recent works in few-shot node classification [8, 10–12, 34–39] successfully transfer the idea to the graph domain. It works as follows: during the training phase, it generates a number of meta-train tasks (or episodes) $\mathcal{T}_{tr}$ from $\mathbb{C}_{base}$ to emulate the test tasks, following their $N$-way $K$-shot node classification specifications:

$$
\begin{aligned}
\mathcal{T}_{tr} &= \{\mathcal{T}_t\}_{t=1}^{T} = \{\mathcal{T}_1, \mathcal{T}_2, ..., \mathcal{T}_T\}, \\
\mathcal{T}_t &= \{\mathcal{S}_t, \mathcal{Q}_t\}, \\
\mathcal{S}_t &= \{(v_1, y_1), (v_2, y_2), ..., (v_{N \times K}, y_{N \times K})\}, \\
\mathcal{Q}_t &= \{(v_1, y_1), (v_2, y_2), ..., (v_{N \times K}, y_{N \times K})\}.
\end{aligned}
\tag{1}
$$

For a typical meta-learning based method, in each episode, $K$ labeled nodes are randomly sampled from $N$ base classes, forming a *support set*, to train the GNN model while emulating the $N$-way $K$-shot node classification in the test phase. Then GNN predicts labels for an emulated *query set* of nodes randomly sampled from the same classes as the support set. The Cross-Entropy Loss ($L_{CE}$) is calculated to optimize the GNN encoder $g_\theta$ and the classifier $f_\psi$ in an end-to-end fashion:

$$
\theta, \psi = \arg \min_{\theta, \psi} L_{CE}(\mathcal{T}_t; \theta, \psi).
\tag{2}
$$

Based on this, Meta-GNN [34] combines MAML [31] with GNNs to achieve optimization for different meta-tasks. GPN [8] applies ProtoNet [30] and computes node importance for a transferable metric function. G-Meta [10] aims to establish a local subgraph for each node to achieve fast adaptations to new meta-tasks. RALE [35] obtains relative and absolute node embeddings based on node positions on graphs to model node dependencies in each meta-task. An exhaustive survey is beyond the scope of this paper; see [13] for an overview. However, all those methods are evaluated on different datasets with each own evaluation protocol, which fragments the practical knowledge on how meta-learning performs with a few labeled nodes and makes it hard to explicitly compare their superiority or inferiority. To bridge this gap, in this paper, we conduct extensive experiments to compare new advances and prior works for FSNC tasks uniformly and comprehensively.

## 2.3 A Motivating Example and Preliminary Analysis

More recently, related works in the image domain demonstrate that the reason for the fast adaptation lies in feature reuse rather than those complicated mate-learning algorithms [15, 16]. In other words, with a carefully pretrained encoder, decent performance can be obtained through directly fine-tuning a simple classifier on the target task. However, few studies have been done on the graph domain due to its important difference from images that nodes in a graph are not i.i.d. Their interactive relationships are reflected by both the topological and semantic information. To validate such hypothesis on graphs, based on [16], we construct an *Intransigent GNN* model, namely *I-GNN*, that simply does not adapt to new tasks. We decouple the training procedure to two separate phases. In the first phase, a GNN encoder $g_\theta$ with a linear classifier $f_\phi$ as the classifier is simply pretrained on all base classes $\mathbb{C}_{base}$ with vanilla supervision through $L_{CE}$:

$$\mathcal{T}'_{tr} = \cup\{\mathcal{T}_t\}_{t=1}^T = \cup\{\mathcal{T}_1, \mathcal{T}_2, ..., \mathcal{T}_T\},$$
$$\theta, \phi = \arg\min_{\theta,\phi} L_{CE}(\mathcal{T}'_{tr}; \theta, \phi) + \mathcal{R}(\theta), \tag{3}$$

where $\mathcal{R}(\theta)$ is a weight-decay regularization term: $\mathcal{R}(\theta) = \|\theta\|^2/2$. Then, we freeze the parameter of the GNN encoder $g_\theta$ and discard the classifier $f_\phi$. When fine-tuning on a target few-shot node classification task $\mathcal{T}_i = \{\mathcal{S}_i, \mathcal{Q}_i\}$, the embeddings of all nodes from $\mathcal{T}_i$ are directly transferred from the pretrained GNN encoder $g_\theta$. Then another linear classifier $f_\psi$ is involved and tuned with few-shot labeled nodes from the support set $\mathcal{S}_i$ to predict labels of nodes in the query set $\mathcal{Q}_i$:

$$\psi = \arg\min_\psi L_{CE}(\mathcal{S}_i; \theta, \psi). \tag{4}$$

**Results and Analysis of the Intransigent GNN model I-GNN.** We demonstrate the performance of the intransigent model and compare it with those meta-learning based models in Table 1, 6. Under the same evaluation protocol (defined in Section 3.2), the simple intransigent model I-GNN has very competitive performance with meta-learning based methods. On datasets (e.g., CiteSeer) where the number of base classes $|\mathbb{C}_{base}|$ is limited, I-GNN consistently outperforms meta-learning based methods in terms of accuracy. This motivating example concludes that transferring node embeddings from the vanilla supervised training method I-GNN could be an alternative to meta-learning. Moreover, we take one step further and postulate that if more transferable node embeddings are obtained during pretraining, the performance on target FSNC tasks could be improved even more.

## 2.4 Transductive Linear Probing for Few-shot Node Classification.

Inspired by the motivating example above, we generalize it to a new framework, *Transductive Linear Probing* (TLP), for few-shot node classification. The only difference between TLP and I-GNN is that the pretraining method can be an arbitrary strategy rather than the vanilla supervised learning. It can even be self-supervised training methods that do not have any requirement on base classes. In this way, the second line of Eq. (3) can be generalized to:

$$\theta = \arg\min_\theta L_{pretrain}(\mathcal{T}'_{tr}; \theta), \tag{5}$$

where $L_{pretrain}$ is an arbitrary loss function to pretrain the GNN encoder $g_\theta$. Then following Eq. (4), we can exploit a linear classifier to probe the transferred embeddings of nodes from novel classes, and perform the final node classification.

In this paper, we thoroughly investigate Graph Contrastive Learning (GCL) as the pretraining strategy for TLP due to two reasons: (1) GCL [17, 19, 20, 23, 40, 41] is a proved effective way to learn generalizable node representations in either a supervised or self-supervised manner. By maximizing the consistency over differently transformed positive and negative examples (termed as views), GCL enforces the GNNs to be aware of the semantic and topological knowledge and injected perturbations on graphs. Trained on the global structures, GCL should be capable of addressing the piecemeal knowledge issue in meta-learning to increase the generalizability of the learned GNNs. Also, [42] summarizes the characteristics of GCL frameworks and empirically demonstrates the transferability of the learned representations. (2) GCL has no requirement for the base classes, which means GCL can be deployed even when the number of base classes is limited, or the nodes in base classes are unlabeled. The effectiveness of GCL highly relies on the contrastive loss function. There are two categories of contrastive loss function for graphs: (1) Supervised Contrastive

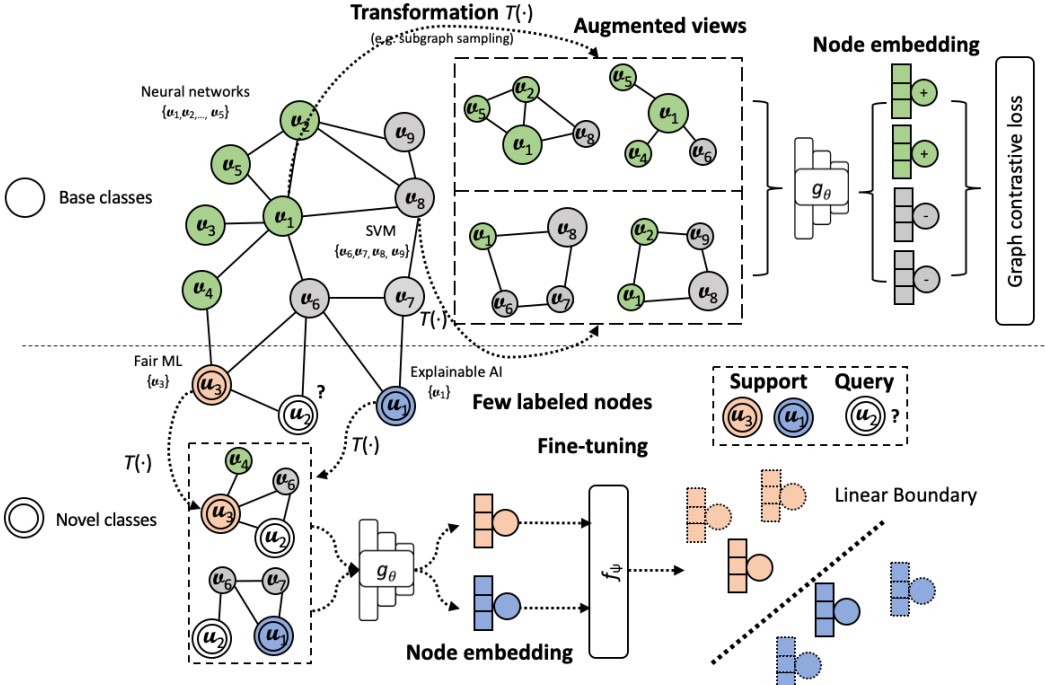

**Figure 1:** The framework of TLP with supervised GCL: (a) Supervised GCL framework. (b) Fine-tuning on few-shot labeled nodes from novel classes with support and query sets. Colors indicate different classes (e.g., *Neural Networks, SVM, Fair ML, Explainable AI*). Specially, white nodes mean labels of those nodes are unavailable. Labels of all nodes in base classes are available. Different types of nodes indicate if nodes are from base classes or novel classes. The counterpart of TLP with self-supervised GCL is very simliar to this, and a figure is included in Appendix B.

Loss ($L_{SupCon}$) [43, 44]. (2) Self-supervised Contrastive Loss: Information Noise Contrastive Estimation ($L_{InfoNCE}$) [19, 20, 22] and Jensen-Shannon Divergence ($L_{JSD}$) [17, 18]. We also consider a special GCL method, BGRL [21], which does not explicitly require negative examples. The framework for TLP with an iconic supervised GCL method is provided in Fig. 1. From another perspective, our work is the first to focus on the extrapolation ability of GCL methods, especially under extremer few-shot settings without labels for nodes in base classes.

## 3 Experimental Study

### 3.1 Experimental Settings

We conduct systematic experiments to compare the performance of meta-learning and TLP methods (with self-supervised and supervised GCL) on the few-shot node classification task. For meta-learning, we evaluate **ProtoNet** [30], **MAML** [31], **Meta-GNN** [34], **G-Meta** [10], **GPN** [8], **AMM-GNN** [9], and **TENT** [12]. For TLP methods with both self-supervised and supervised forms, we evaluate **MVGRL** [17], **GraphCL** [18], **GRACE** [19], **MERIT** [20], and **SUGRL** [22]. Moreover, **BGRL** [45] and **I-GNN** [16] are exclusively used for TLP methods with self-supervised GCL or supervised GCL, respectively. The detailed descriptions of these models can be found in Appendix D. For comprehensive studies, we benchmark those methods on six prevalent real-world graph datasets: CoraFull [46], ogbn-arxiv [47], Coauthor-CS [48], Amazon-Computer [48], Cora [49], and CiteSeer [49]. Specifically, each dataset is a connected graph and consists of multiple node classes for training and evaluation. A more detailed description of those datasets is provided in Appendix F with their statistics and class split policies in Table 3 in Appendix E.

## 3.2 Evaluation Protocol

In this section, we specify the evaluation protocol used to compare both meta-learning based methods and TLP based methods. For an attributed graph dataset $\mathcal{G} = (\mathbf{A}, \mathbf{X})$ with a divided node label space $\mathbb{C} = \{\mathbb{C}_{base}, \mathbb{C}_{novel} \text{ (or } \mathbb{C}_{test})\}$, we split $\mathbb{C}_{base}$ into $\mathbb{C}_{train}$ and $\mathbb{C}_{dev}$ (The split policy for each datasets are listed in Table 3). For evaluation, given a GNN encoder $g_\theta$, a classifier $f_\psi$, the validation epoch interval $V$, the number of sampled meta-tasks for evaluation $I$, the epoch patience $P$, the maximum epoch number $E$, the experiment repeated times $R$, and the $N$-way, $K$-shot, $M$-query setting specification, the final FSNC accuracy $\mathcal{A}$ and the confident interval $\mathcal{I}$ (two mainly-concerned metrics) are calculated according to Algorithm 1 given below. The default values of all those parameters are given in Table 2 in Appendix C.

---

**Algorithm 1** UNIFIED EVALUATION PROTOCOL FOR FEW-SHOT NODE CLASSIFICATION

---

**Input:** Graph $\mathcal{G}$, $\mathbb{C}_{train}$, $\mathbb{C}_{dev}$, $\mathbb{C}_{test}$; GNN $g_\theta$, classifier $f_\psi$; parameters $V$, $I$, $P$, $E$, $R$, $N$, $K$, $M$
**Output:** Trained models $g_\theta$ and $f_\psi$, accuracy $\mathcal{A}$, confident interval $\mathcal{I}$.
    // Repeat experiment for $R$ times
1: **for** $r = 1, 2, \ldots, R$ **do**
2:     $p \leftarrow 1, t \leftarrow 1, s_{best} \leftarrow 0$;
3:     **while** $t \leq E$ **do**
4:         Optimize $g_\theta$ based on the specific training strategy (i.e., meta-learning and TLP);     // Training
5:         **if** $t \mod V = 0$ **then**
6:             Sample $I$ meta-tasks from $\mathbb{C}_{dev}$ on $\mathcal{G}$;     // Validation
7:             Calculate the obtained few-shot node classification accuracy $s$;
8:             **if** $s > s_{best}$ **then**
9:                 $s_{best} \leftarrow s, p \leftarrow 0$;
10:            **else**
11:                $p \leftarrow p + 1$;
12:            **end if**
13:         **end if**
14:         **if** $p = P$ **then**
15:             **break**;     // Early Break
16:         **end if**
17:     **end while**
18:     Sample $I$ meta-tasks from $\mathbb{C}_{test}$ on $\mathcal{G}$;     // Test
19:     Calculate the obtained classification accuracy $s_{test}$;
20:     $s_r \leftarrow s_{test}, r \leftarrow r + 1$;
21: **end for**
22: Calculate averaged accuracy $\mathcal{A}$ and confident interval $\mathcal{I}$ based on $\{s_1, s_2, \ldots, s_r\}$;

---

## 3.3 Comparison

Table 1 presents the performance comparison of all methods on the few-shot node classification task. Specifically, we give results under four different few-shot settings to exhibit a more comprehensive comparison: 5-way 1-shot, 5-way 5-shot, 2-way 1-shot, and 2-way 5-shot. More results are given in Appendix H. We choose the average classification accuracy and the 95% confidence interval over $R$ repetitions as the evaluation metrics. From Table 1, we discover the following observations:

- TLP methods **consistently outperforms** meta-learning methods, which indicates the importance of transferring comprehensive node representations in FSNC tasks. In TLP methods, the model is forced to extract node-level structural information, while the meta-learning methods mainly focus on label information. As a result, TLP methods can transfer better node representations and exhibit superior performance on meta-test tasks.

- Even **without using any label information** from base classes, TLP with self-supervised GCL methods can mostly outperform TLP with supervised GCL methods. This signifies that directly injecting supervision can potentially hinder the generalizability for TLP, which is further investigated in the following sections.

- **Increasing the number of shots** $K$ (i.e., number of labeled nodes in the support set) has more significant effect on performance of both forms of TLP methods, compared with meta-learning methods. This is due to the fact that with the additional support nodes, TLP with GCL can provide more informative node representations to learn a more powerful classifier. Instead, the

**Table 1:** The overall few-shot node classification results of meta-learning methods and TLP with various GCL methods under different settings. Accuracy ($\uparrow$) and confident interval ($\downarrow$) are in %. The best and second best results are **bold** and underlined, respectively. OOM denotes out of memory.

| Dataset | CoraFull | | ogbn-arxiv | | CiteSeer | |
|---|---|---|---|---|---|---|
| Setting | 5-way 1-shot | 5-way 5-shot | 5-way 1-shot | 5-way 5-shot | 2-way 1-shot | 2-way 5-shot |
| Meta-learning | | | | | | |
| MAML [31] | $22.63 \pm 1.19$ | $27.21 \pm 1.32$ | $27.36 \pm 1.48$ | $29.09 \pm 1.62$ | $52.39 \pm 2.20$ | $54.13 \pm 2.18$ |
| ProtoNet [30] | $32.43 \pm 1.61$ | $51.54 \pm 1.68$ | $37.30 \pm 2.00$ | $53.31 \pm 1.71$ | $52.51 \pm 2.44$ | $55.69 \pm 2.27$ |
| Meta-GNN [34] | $55.33 \pm 2.43$ | $70.50 \pm 2.02$ | $27.14 \pm 1.94$ | $31.52 \pm 1.71$ | $56.14 \pm 2.62$ | $67.34 \pm 2.10$ |
| GPN [8] | $52.75 \pm 2.32$ | $72.82 \pm 1.88$ | $37.81 \pm 2.34$ | $50.50 \pm 2.13$ | $53.10 \pm 2.39$ | $63.09 \pm 2.50$ |
| AMM-GNN [9] | $58.77 \pm 2.49$ | $75.61 \pm 1.78$ | $33.92 \pm 1.80$ | $48.94 \pm 1.87$ | $54.53 \pm 2.51$ | $62.93 \pm 2.42$ |
| G-Meta [10] | **$60.44 \pm 2.48$** | **$75.84 \pm 1.70$** | $31.48 \pm 1.70$ | $47.16 \pm 1.73$ | $55.15 \pm 2.68$ | $64.53 \pm 2.35$ |
| TENT [12] | $55.44 \pm 2.08$ | $70.10 \pm 1.73$ | **$48.26 \pm 1.73$** | **$61.38 \pm 1.72$** | **$62.75 \pm 3.23$** | **$72.95 \pm 2.13$** |
| TLP with Supervised GCL | | | | | | |
| I-GNN [16] | $42.70 \pm 1.92$ | $51.46 \pm 1.69$ | $38.46 \pm 1.77$ | $51.46 \pm 1.69$ | $58.70 \pm 3.17$ | $65.60 \pm 2.58$ |
| MVGRL [17] | $44.98 \pm 1.99$ | $71.18 \pm 1.75$ | OOM | OOM | $55.79 \pm 1.39$ | $66.72 \pm 2.13$ |
| GraphCL [18] | $47.00 \pm 1.64$ | $67.94 \pm 1.71$ | OOM | OOM | $53.55 \pm 1.68$ | $69.50 \pm 1.41$ |
| GRACE [19] | **$65.48 \pm 2.45$** | **$85.08 \pm 1.49$** | OOM | OOM | $61.20 \pm 2.39$ | **$81.76 \pm 1.74$** |
| MERIT [20] | $52.80 \pm 2.72$ | $81.30 \pm 1.53$ | OOM | OOM | $61.25 \pm 2.59$ | $81.45 \pm 1.80$ |
| SUGRL [22] | $54.26 \pm 2.24$ | $77.55 \pm 1.95$ | **$52.13 \pm 2.11$** | **$70.05 \pm 1.56$** | **$65.34 \pm 2.55$** | $75.81 \pm 1.43$ |
| TLP with Self-supervised GCL | | | | | | |
| MVGRL [17] | $59.91 \pm 2.39$ | $76.76 \pm 1.63$ | OOM | OOM | $64.45 \pm 2.77$ | $80.25 \pm 1.82$ |
| GraphCL [18] | $64.20 \pm 2.56$ | $83.74 \pm 1.46$ | OOM | OOM | $73.55 \pm 3.09$ | **$92.35 \pm 1.24$** |
| BGRL [45] | $43.83 \pm 2.11$ | $70.44 \pm 1.62$ | $36.76 \pm 1.74$ | $53.44 \pm 0.36$ | $54.32 \pm 1.63$ | $70.50 \pm 2.11$ |
| GRACE [19] | $72.42 \pm 2.06$ | $83.82 \pm 1.67$ | OOM | OOM | $60.75 \pm 2.54$ | $78.42 \pm 2.01$ |
| MERIT [20] | $73.38 \pm 2.25$ | **$87.66 \pm 1.43$** | OOM | OOM | $64.53 \pm 2.81$ | $90.32 \pm 1.66$ |
| SUGRL [22] | **$77.35 \pm 2.20$** | $83.96 \pm 1.52$ | **$60.04 \pm 2.11$** | **$77.52 \pm 1.45$** | **$77.34 \pm 2.83$** | $86.32 \pm 1.57$ |

meta-learning methods are based on the extracted label information and thus cannot benefit from additional node-level information.

- Most TLP methods encounter the **OOM (out of memory) problem** when applied to the `ogbn-arxiv` dataset. This is due to the fact that the contrastive strategy in TLP methods will consume a larger memory compared with traditional supervised learning. Thus, the scalability problem is not negligible for TLP with GCL methods.

- BGRL [45] exhibits **less competitive** performance compared with other TLP methods with self-supervised GCL. The result indicates that negative samples are important for self-supervised GCL in FSNC, which can help the model exploit node-level information. Nevertheless, without the requirement of negative samples, BGRL can parallel better to handle the OOM problem.

## 3.4 Further Analysis

To explicitly compare the results between meta-learning and TLP and between two forms of TLP, we provide further results of all methods on various $N$-way $K$-shot settings in Fig. 2 and Fig. 3. From the results, we can obtain the following observations:

- When a larger values of $N$ is presented, the performance drop is **less significant** on TLP based methods compared to meta-learning based methods. The performance of all methods degrades as $N$ increases (i.e., more classes in each meta-task). With a larger $N$, the variety of classes in each meta-task can result in a more complex class distribution and thus increase the classification difficulties. Nevertheless, the performance drop is less significant on TLP with both forms of GCL methods. This is because the utilized GCL methods focus more on node-level structural patterns, which incorporate more potentially useful information for classification. As a result, TLP is more capable of alleviating the problem of difficult classification caused by a larger $N$.

- As shown in Fig. 3, the **performance improvement** of TLP with self-supervised GCL methods over meta-learning methods on `CiteSeer` is generally more impressive than other datasets. The main reason is that `CiteSeer` bears a significantly smaller class set (2/2/2 classes for $\mathbb{C}_{train}$/$\mathbb{C}_{dev}$/$\mathbb{C}_{test}$). In consequence, the meta-learning methods cannot effectively leverage the supervision information during training. Nevertheless, TLP with self-supervised GCL can extract useful structural information for better generalization performance.

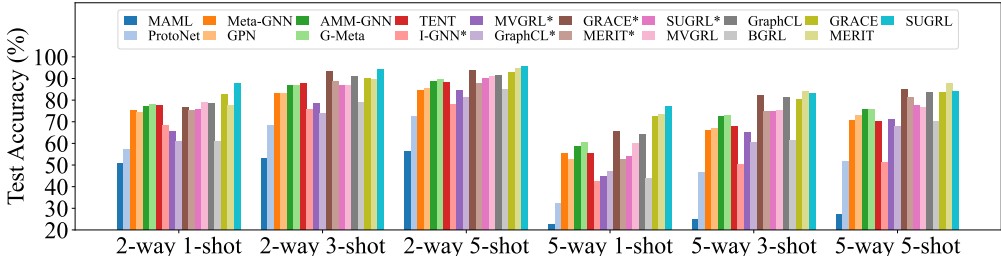

**Figure 2:** $N$-way $K$-shot results on `CoraFull`, meta-learning and TLP. TLP Methods with $*$ are based on supervised GCL methods and I-GNN.

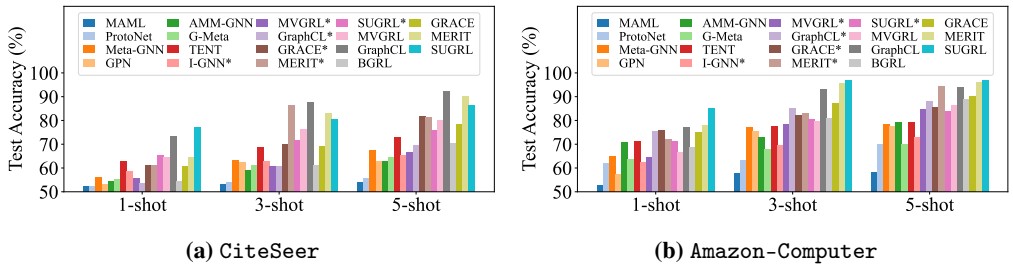

**Figure 3:** 2-way $K$-shot results on `CiteSeer` and `Amazon-Computer`, meta-learning and two forms of TLP. TLP Methods with $*$ are based on supervised GCL methods and I-GNN.

### 3.5 Effect of Supervision Information in Base Classes

In this section, we further investigate the effectiveness of the supervised information in TLP with supervised GCL methods. Specifically, we leverage a combined loss $L_{JointCon} = \lambda L_{SelfCon} + (1 - \lambda)L_{SupCon}$, where $L_{SelfCon}$ indicates a self-supervised GCL loss, either $L_{JSD}$ or $L_{InfoNCE}$ according to the models, and $L_{JointCon}$ is a mixture of supervised GCL loss and self-supervised GCL loss. In this way, we can gradually adjust the value of $\lambda$ to inject different levels of supervision signals into GCL and then observe the performance fluctuation. Note that due to the unstable training curve brought by the joint loss $L_{JointCon}$, we increase the epoch patience number from $P$ to $2P$ to ensure convergence. The results on `Cora` dataset (we observe similar results on other datasets) with different values of $\lambda$ are provided in Fig. 4. From the results, we can obtain the following observations:

- In general, the classification performance **increases with a larger value of** $\lambda$. In other words, directly injecting supervision information into GCL for TLP will usually reduce the performance on few-shot node classification tasks. Nevertheless, carefully injecting supervision information can slightly increase the accuracy by choosing a suitable value of $\lambda$. On the other hand, the results also verify that the TLP framework can still achieve considerable performance without any explicit restrictions for base classes.

- Even with **a relatively small value of** $\lambda$ (e.g., 0.1), the performance improvement over TLP with totally supervised GCL (i.e., $\lambda = 0.0$) is still significant. That being said, the contrastive strategy that leverages graph structures can provide better performance by providing comprehensive node representations.

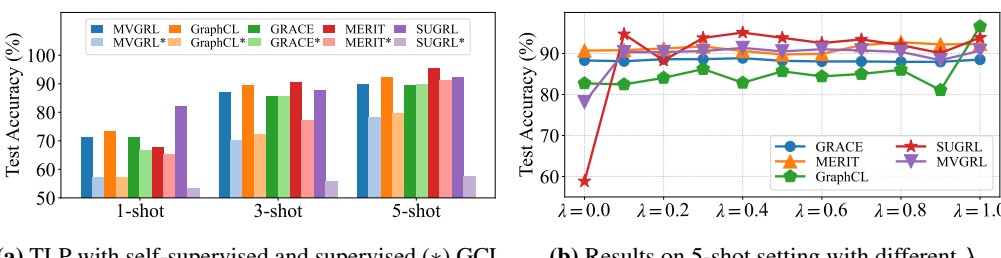

**(a)** TLP with self-supervised and supervised (∗) GCL     **(b)** Results on 5-shot setting with different $\lambda$

**Figure 4:** Results on dataset `Cora` (2-way)

## 3.6 Evaluating Learned Node Representations on Novel Classes

In this section, we further validate the quality of the learned node representations from different training strategies. Particularly, we leverage two prevalent clustering evaluation metrics: *normalized mutual information* (NMI) and *adjusted random index* (ARI), on learned node representations clustered based on K-Means. We evaluate the representations learned from two datasets `CoraFull` and `CiteSeer` for a fair comparison. The results are presented in Table 4 in Appendix H.2 . Based on the results, we can obtain the following observations:

- The meta-learning methods typically exhibit **inferior NMI and ARI scores** compared with both forms of TLP. This is because meta-learning methods are dedicated for extracting supervision information from node samples and thus cannot fully utilize node-level structural information.

- In general, TLP with self-supervised GCL methods can result in **larger values of both NMI and ARI scores** than TLP with supervised GCL. This is due to the fact that the self-supervised GCL model focuses more on extracting structural information without the interruption of label information. As a result, the learned node representations are more comprehensive and thus exhibit superior clustering performance.

- **The difference of NMI and ARI score**s between meta-learning and TLP is more significant on `CiteSeer` than `CoraFull`. This phenomenon potentially results from the fact that `CiteSeer` consists of fundamentally fewer classes than `CoraFull`. In consequence, for `CiteSeer`, the meta-learning methods will largely rely on label information instead of node-level structural information for classification.

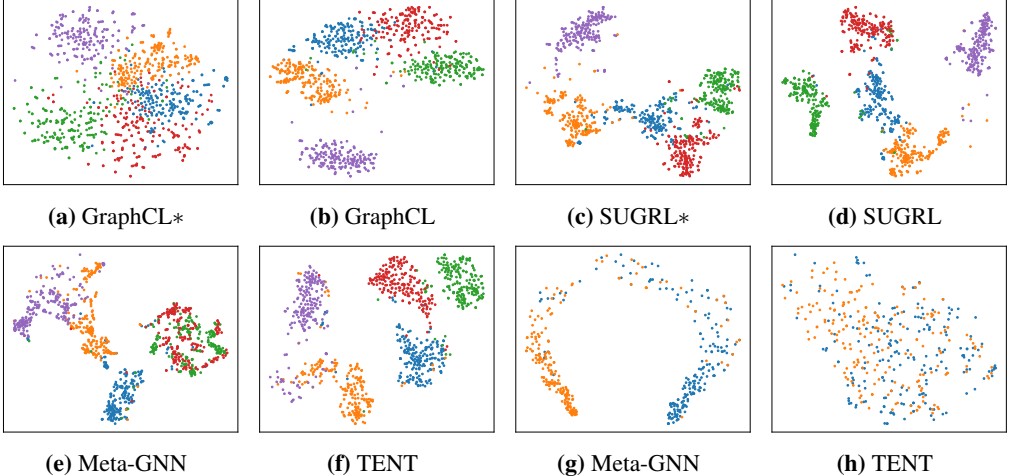

**(a)** GraphCL∗     **(b)** GraphCL     **(c)** SUGRL∗     **(d)** SUGRL

**(e)** Meta-GNN     **(f)** TENT     **(g)** Meta-GNN     **(h)** TENT

**Figure 5:** The t-SNE visualization results. Fig. (a)-(f) are for dataset `CoraFull` (5-way). Fig. (g)-(h) are for dataset `CiteSeer` (2-way). TLP methods with ∗ are based on supervised GCL methods.

### 3.7 Visualization

To provide an explicit comparison of different baselines, we visualize the learned node representations from `CoraFull` and `CiteSeer` via the t-SNE algorithm, where colors denote different classes. It is noteworthy that for clarity, we randomly select five classes from $\mathbb{C}_{test}$ for the visualization. The results are provided in Fig. 5 (more results are included in Fig. 8). Specifically, we discover that:

- TLP with **self-supervised GCL** generally outperforms TLP with supervised GCL. This is because without learning label information, TLP with self-supervised GCL can concentrate on node representation patterns, which are easier to transfer to target unseen novel classes.

- The learned node representations are **less discriminative** for meta-learning on `CiteSeer` compared with `CoraFull`. This is because `CiteSeer` contains fewer classes, which means the node representations learned by meta-learning methods will be less informative, since they are only required to classify nodes from a small class set.

## 4  Conclusion, Limitations, and Outlook

In this paper, we propose TLP as an alternative framework to meta-learning for FSNC tasks. First, we provide a motivating example, a vanilla intransient GNN model, to validate our postulation that a generalizable GNN encoder is the key to FSNC tasks. Then, we provide a formal definition for TLP, which transfers node embeddings from GCL pretraining to the prevailing meta-learning paradigm. We conduct comprehensive experiments and compare various meta-learning based and TLP-based methods under the same protocol. Our rigorous empirical study reveals several interesting findings on the strengths and weaknesses of the two approaches and identifies that adaptability and scalability are the promising directions for meta-learning based and TLP-based methods, respectively.

However, due to limited space, several limitations of our work need to be acknowledged.

- **Limited design considerations.** Even though an exhaustive survey on FSNC or GCL is out of the scope of this work, we do not provide a more fine-grained comparison on model details, such as different GNN encoders or various transformations during GCL pretraining. Also, we only consider methods applied on a single graph, which currently are the mainstream of research on FSNC. There are more recent works (e.g., [50]) studying FSNC across multiple graphs.

- **Lack of theoretical justifications.** Our findings are based on empirical studies, which cannot disclose the underlying mathematical mechanisms of those methods, such as the performance guarantee by transferring node embeddings from different GCL methods.

How to address these limitations is saved as future work. Note that based on the experiments here, the observations drawn are not conclusive termination. We only cover existing methods in this work and hope this work to be inspiring for developing meta-learning based FSNC methods that can outperform TLP based methods, or better ways to utilize labels in TLP methods. In broader terms, this work lies at the confluence of graph few-shot learning and graph contrastive learning. We hope this work can facilitate the sharing of insights for both communities. On the one hand, we hope our work provides a necessary yardstick to measure progress across the FSNC field. On the other hand, our work should have exhibited several practical guidelines for future research in both vigorous fields. For example, the meta-learning community can get inspired by GCL to learn more transferable graph patterns. Also, few-shot TLP can serve as a new metric to evaluate the extrapolation ability of GCL methods.

### Acknowledgements

This work is supported by the National Science Foundation under grants IIS-2006844, IIS-2144209, IIS-2223769, IIS-2229461, CNS-2154962, and BCS-2228534, the Army Research Office (ARO) W911NF2110030, the Office of Naval Research N00014-21-1-4002, the JP Morgan Chase Faculty Research Award, the Cisco Faculty Research Award, and the Jefferson Lab subcontract JSA-22-D0311.

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

# A    Framework for Meta-learning Based FSNC Methods

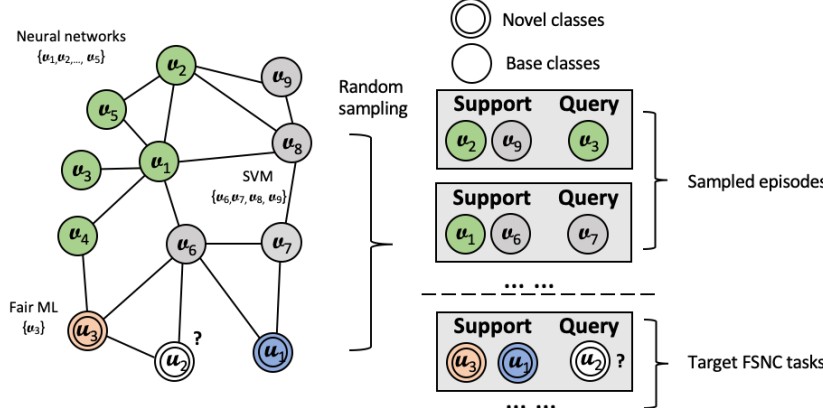

**Figure 6:** The framework for meta-learning methods. Colors indicate different classes (e.g., *Neural Networks, SVM, Fair ML, Explainable AI*). Specifically, white nodes denotes that the labels of those nodes are unavailable. Labels of all nodes in base classes are available. Different types of nodes indicate if nodes are from base classes or novel classes.

# B    Framework for TLP with Self-Supervised GCL

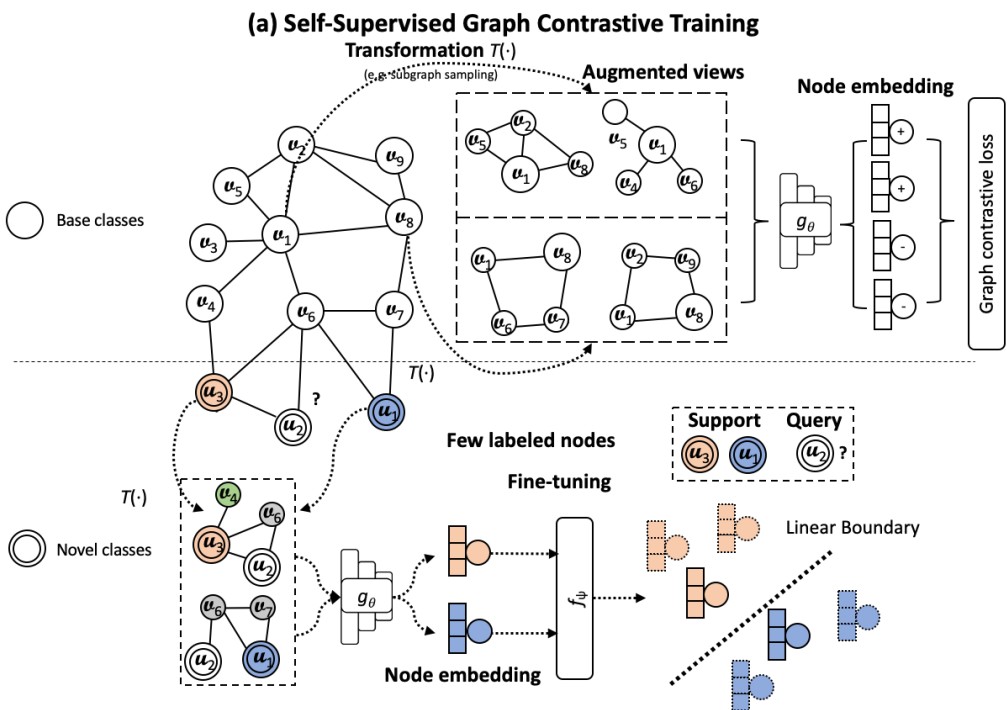

**Figure 7:** The framework for TLP with self-supervised methods. Labels of all nodes in base classes are unavailable. Different types of nodes indicate if nodes are from base classes or novel classes.

## C  Default Values of Parameters in Evaluation Protocol

In this section, we provide the default values of parameters used in our experiments. The details are provided in Table 2. It is noteworthy that the parameters are consistent for all models in both meta-learning and TLP methods. For the experiments that utilize a joint loss of TLP with self-supervised GCL and supervised GCL, we increase the patience number from $P$ to $2P$ to ensure convergence.

**Table 2:** Default Values of Parameters in Evaluation Protocol for Experiments

| Parameters | Description | Value |
|:---:|:---:|:---:|
| $V$ | validation epoch interval | 10 |
| $I$ | number of sampled meta-tasks for evaluation | 100 |
| $P$ | patience number | 10 |
| $E$ | maximum epoch number | 10000 |
| $R$ | number of repeated experiments | 5 |
| $N$ | number of classes in each meta-task | 2,5 |
| $K$ | number of nodes for each class in each meta-task | 1,3,5 |
| $M$ | number of queries for each class in each meta-task | 10 |

## D  Description of Baselines

In this section, we provide further details about the baselines used in our experiments.

*Meta-learning* based methods:

- **ProtoNet** [30]: ProtoNet learns a prototype for each class in meta-tasks by averaging the embeddings of samples in this class. Then it conducts classification on query instances based on their distances to prototypes.

- **MAML** [31]: MAML first optimizes model parameters according to the gradients calculated on the support instances for several steps. Then it meta-updates parameters based on the loss of query instances calculated with the parameters updated on support instances.

- **Meta-GNN** [34]: Meta-GNN combines GNNs with the MAML strategy to apply meta-learning on graph-structured data. Specifically, Meta-GNN learns node embeddings with GNNs, while updating and meta-updating the GNN parameters based on the MAML strategy.

- **G-Meta** [10]: G-Meta extracts a subgraph for each node to learn the node representation with GNNs. Then it conducts the classification on query nodes based on the MAML strategy to update and meta-update the parameters of GNNs.

- **GPN** [8]: GPN proposes to learn node importance for each node in meta-tasks to select more beneficial nodes for classification. Then GPN utilizes ProtoNet to learn node prototypes via averaging node embeddings in a weighted manner.

- **AMM-GNN** [9]: AMM-GNN proposes to extend MAML with an attribute matching mechanism. Specifically, the node embeddings will be adjusted according to the embeddings of nodes in the entire meta-task in an adaptive manner.

- **TENT** [12]: TENT reduces the variance among different meta-tasks for better generalization performance. In particular, TENT learns node and class representations by conducting node-level and class-level adaptations. It also incorporates task-level adaptations that maximizes the mutual information between the support set and the query set.

*Transductive Linear Probing* with different Pretraining methods:

- **I-GNN** [16]: I-GNN learns a GNN encoder with a classifier that is trained on all base classes $\mathbb{C}_{base}$ with the vanilla Cross-Entropy loss $L_{CE}$. Then for each meta-test task, the GNN will be frozen and a new classifier is learned based on the support set for classification.

- **MVGRL** [17]: MVGRL learns node and graph level representations by contrasting the representations of two structural views of graphs, which include first-order neighbors and a graph diffusion. It utilizes a Jensen-Shannon Divergence based contrastive loss $L_{JSD}$.

- **GraphCL** [18]: GraphCL proposes to leverage combinations of different transformations in GCL to facilitate GNNs with generalizability, transferrability, and robustness without sophisticated architectures. It also uses $L_{JSD}$ as the objective.

- **GRACE** [19]: GRACE proposes a hybrid scheme for generating different graph views on both structure and attribute levels. GRACE further provides theoretical justifications behind the motivation. It proposes a variant of Information Noise Contrastive Estimation $L_{InfoNCE}$ as the contrastive loss.

- **MERIT** [20]: MERIT employs two different objectives named cross-view and cross-network contrastiveness to further maximize the agreement between node representations across different views and networks. It uses $L_{InfoNCE}$ similar to that in GRACE as the loss function.

- **SUGRL** [22]: SUGRL proposes to simultaneously enlarge inter-class variation and reduce intra-class variation. The experimental results show promising improvements of generalization error with SUGRL. It also uses $L_{InfoNCE}$ similar to that in GRACE as the loss function.

- **BGRL** [45]: BGRL leverages the concept of BYOL [51] and applies it to graph-structured data by enforcing the agreement between positive views without any explicitly designs on negative views. Specially, it uses Mean Squared Error $L_{MSE}$ between positive views as the final loss.

## E    Statistics of Benchmark Datasets

**Table 3:** Statistics of node classification datasets.

| Dataset | # Nodes | # Edges | # Features | $|\mathbb{C}|$ | $|\mathbb{C}_{train}|$ | $|\mathbb{C}_{dev}|$ | $|\mathbb{C}_{test}|$ |
|---|---|---|---|---|---|---|---|
| CoraFull | 19,793 | 63,421 | 8,710 | 70 | 40 | 15 | 15 |
| ogbn-arxiv | 169,343 | 1,166,243 | 128 | 40 | 20 | 10 | 10 |
| Coauthor-CS | 18,333 | 81,894 | 6,805 | 15 | 5 | 5 | 5 |
| Amazon-Computer | 13,752 | 245,861 | 767 | 10 | 4 | 3 | 3 |
| Cora | 2,708 | 5,278 | 1,433 | 7 | 3 | 2 | 2 |
| CiteSeer | 3,327 | 4,552 | 3,703 | 6 | 2 | 2 | 2 |

## F    Description of Benchmark Datasets

In this section, we provide the detailed descriptions of the benchmark datasets used in our experiments. All the datasets are public and available on both PyTorch-Geometric [52] and DGL [53].

- **CoraFull** [46] is a citation network that extends the prevalent small cora network. Specifically, it is achieved from the entire citation network, where nodes are papers, and edges denote the citation relations. The classes of nodes are obtained based on the paper topic. For this dataset, we use 40/15/15 node classes for $\mathbb{C}_{train}$/$\mathbb{C}_{dev}$/$\mathbb{C}_{test}$.

- **ogbn-arxiv** [47] is a directed citation network that consists of CS papers from MAG [54]. Here nodes represent CS arXiv papers, and edges denote the citation relations. The classes of nodes are assigned based on the 40 subject areas of CS papers in arXiv. For this dataset, we use 20/10/10 node classes for $\mathbb{C}_{train}$/$\mathbb{C}_{dev}$/$\mathbb{C}_{test}$.

- **Coauthor-CS** [48] is a co-authorship graph based on the Microsoft Academic Graph from the KDD Cup 2016 challenge. Here, nodes are authors, and are connected by an edge if they co-authored a paper; node features represent paper keywords for each author's papers, and class labels indicate most active fields of study for each author. For this dataset, we use 5/5/5 node classes for $\mathbb{C}_{train}$/$\mathbb{C}_{dev}$/$\mathbb{C}_{test}$.

- **Amazon-Computer** [48] includes segments of the Amazon co-purchase graph [55], where nodes represent goods, edges indicate that two goods are frequently bought together, node features are bag-of-words encoded product reviews, and class labels are given by the product category. For this dataset, we use 4/3/3 node classes for $\mathbb{C}_{train}/\mathbb{C}_{dev}/\mathbb{C}_{test}$.

- **Cora** [49] is a citation network dataset where nodes mean paper and edges mean citation relationships. Each node has a predefined feature with 1,433 dimensions. The dataset is designed for the node classification task. The task is to predict the category of certain paper. For this dataset, we use 3/2/2 node classes for $\mathbb{C}_{train}/\mathbb{C}_{dev}/\mathbb{C}_{test}$.

- **CiteSeer** [49] is also a citation network dataset where nodes mean scientific publications and edges mean citation relationships. Each node has a predefined feature with 3,703 dimensions. The dataset is designed for the node classification task. The task is to predict the category of certain publication. For this dataset, we use 2/2/2 node classes for $\mathbb{C}_{train}/\mathbb{C}_{dev}/\mathbb{C}_{test}$.

## G  Implementation Details

In this section, we introduce the implementation details for all methods compared in our experiments. Specifically, for the encoders used in TLP methods, we follow the settings in the original papers of the corresponding models to ensure consistency, and we choose *Logistic Regression* as the linear classifier for the final classification. For encoders in meta-learning methods, we utilize the original designs for papers using GNNs. For papers without using GNNs (i.e., ProtoNet [30] and MAML [31]), we use a two-layer GCN [1] as the encoder with a hidden size of 16. We utilize the Adam optimizer [56] for all experiments with a learning rate of 0.001. To effectively initialize the GNNs in our experiments, we leverage the Xavier initialization [57]. For meta-learning methods using the MAML framework, we set the number of meta-update steps as 20 with a meta-learning rate of 0.05. To ensure more stable convergence in meta-learning methods, we set the weight decay rate as $10^{-4}$. We set the dropout rate as 0.5 for better generalization performance. The evaluation protocol parameters are provided in Table 2. All experiments are implemented using PyTorch [58]. We run all experiments on a single 80GB Nvidia A100 GPU.

## H  More Results

### H.1  Visualization

In this section, we provide additional visualization results for more meta-learning and TLP methods on `CoraFull` dataset in Fig. 8.

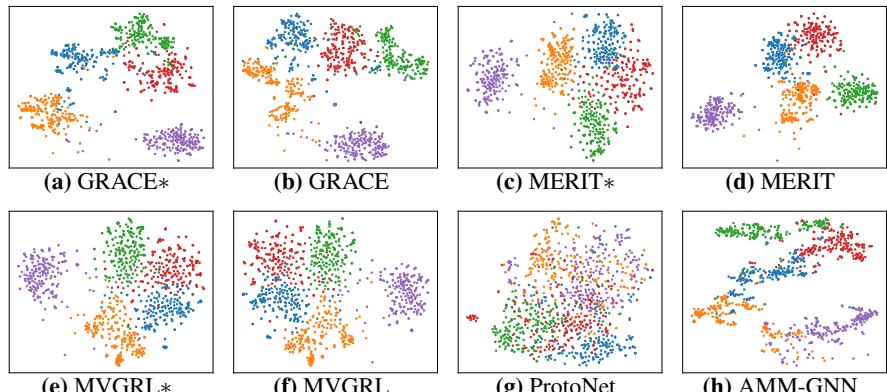

|  |  |  |  |
|---|---|---|---|
| **(a)** GRACE∗ | **(b)** GRACE | **(c)** MERIT∗ | **(d)** MERIT |
| **(e)** MVGRL∗ | **(f)** MVGRL | **(g)** ProtoNet | **(h)** AMM-GNN |

**Figure 8:** The t-SNE visualization results of meta-learning and TLP methods on `CoraFull`. TLP methods with ∗ are based on supervised GCL methods.

.

## H.2 Node Representation Evaluation

In this section, we provide the detailed node representation evaluations on two datasets `CoraFull` and `CiterSeer` based on NMI and ARI scores in Table 4.

**Table 4:** The overall NMI (↑) and ARI (↑) results of meta-learning and TLP methods on two datasets

| Dataset | CoraFull | | CiteSeer | |
|---|---|---|---|---|
| Metrics | NMI | ARI | NMI | ARI |
| Meta-learning | | | | |
| MAML | 0.1622 | 0.0597 | 0.0754 | 0.0602 |
| ProtoNet | 0.2669 | 0.1263 | 0.0915 | 0.0765 |
| AMM-GNN | 0.6247 | 0.5087 | 0.2090 | 0.1781 |
| G-Meta | 0.5003 | 0.3702 | 0.1913 | 0.1502 |
| Meta-GNN | 0.5534 | 0.4196 | 0.1317 | 0.1171 |
| GPN | 0.6001 | 0.4599 | 0.2119 | 0.2087 |
| TENT | 0.5760 | 0.4652 | 0.0930 | 0.0811 |
| Supervised GCL | | | | |
| GRACE | 0.7199 | 0.6239 | 0.4693 | 0.4769 |
| MERIT | 0.6119 | 0.4470 | 0.3471 | 0.3482 |
| GraphCL | 0.2474 | 0.0852 | 0.1321 | 0.0711 |
| SUGRL | 0.7298 | 0.6626 | 0.3927 | 0.4451 |
| MVGRL | 0.6412 | 0.5038 | 0.2445 | 0.2146 |
| Self-supervised GCL | | | | |
| GRACE | 0.6781 | 0.5856 | 0.2663 | 0.2778 |
| MERIT | 0.7419 | 0.6590 | 0.3923 | 0.4014 |
| GraphCL | 0.7023 | 0.5628 | 0.5579 | 0.5890 |
| SUGRL | 0.7680 | 0.7049 | 0.3952 | 0.4460 |
| MVGRL | 0.6227 | 0.4788 | 0.2554 | 0.2232 |

## H.3 Main Results for the Other Three Datasets or Other Settings

In this section, we further provide results for the other three datasets used in our experiments: `Coauthor-CS`, `Amazon-Computer`, and `Cora`, and 2-way classification results on `CoraFull`, `ogbn-arxiv`, and `Coauthor-CS`:

**Table 5:** The overall few-shot node classification results of meta-learning methods and TLP with different GCL methods under different settings. Accuracy (↑) and confidence interval (↓) are in %. The best and second best results are **bold** and underlined, respectively.

| Dataset | Coauthor-CS | | Amazon-Computer | | Cora | |
|---|---|---|---|---|---|---|
| Setting | 5-way 1-shot | 5-way 5-shot | 2-way 1-shot | 2-way 5-shot | 2-way 1-shot | 2-way 5-shot |
| Meta-learning | | | | | | |
| MAML | $27.98 \pm 1.42$ | $42.12 \pm 1.40$ | $52.67 \pm 2.11$ | $58.23 \pm 2.53$ | $53.13 \pm 2.26$ | $57.39 \pm 2.23$ |
| ProtoNet | $32.13 \pm 1.52$ | $49.25 \pm 1.50$ | $61.98 \pm 2.95$ | $70.20 \pm 2.64$ | $53.04 \pm 2.36$ | $57.92 \pm 2.34$ |
| Meta-GNN | $52.86 \pm 2.14$ | $68.59 \pm 1.49$ | $65.19 \pm 3.29$ | $78.65 \pm 3.12$ | $\underline{65.27 \pm 2.93}$ | $72.51 \pm 1.91$ |
| GPN | $60.66 \pm 2.07$ | $\mathbf{81.79 \pm 1.18}$ | $57.26 \pm 1.50$ | $77.63 \pm 2.91$ | $62.61 \pm 2.71$ | $76.39 \pm 2.33$ |
| AMM-GNN | $\underline{62.04 \pm 2.26}$ | $\underline{81.78 \pm 1.24}$ | $\underline{71.04 \pm 3.56}$ | $\underline{79.21 \pm 3.38}$ | $65.23 \pm 2.67$ | $\mathbf{82.30 \pm 2.07}$ |
| G-Meta | $59.68 \pm 2.16$ | $74.18 \pm 1.29$ | $63.68 \pm 3.05$ | $70.21 \pm 3.16$ | $\mathbf{67.03 \pm 3.22}$ | $\underline{80.05 \pm 1.98}$ |
| TENT | $\mathbf{63.70 \pm 1.88}$ | $76.90 \pm 1.19$ | $\mathbf{71.15 \pm 3.11}$ | $\mathbf{79.25 \pm 2.61}$ | $53.05 \pm 2.78$ | $62.15 \pm 2.13$ |
| TLP with Supervised GCL | | | | | | |
| I-GNN | $43.89 \pm 1.82$ | $55.93 \pm 1.46$ | $62.32 \pm 2.89$ | $72.81 \pm 2.93$ | $54.45 \pm 3.13$ | $65.18 \pm 2.21$ |
| MVGRL | $62.16 \pm 2.05$ | $84.79 \pm 1.13$ | $64.69 \pm 2.84$ | $84.84 \pm 2.10$ | $57.24 \pm 2.07$ | $78.04 \pm 2.08$ |
| GraphCL | $54.72 \pm 2.62$ | $84.02 \pm 1.23$ | $\mathbf{75.65 \pm 3.05}$ | $\underline{88.31 \pm 1.86}$ | $57.10 \pm 2.27$ | $79.53 \pm 1.98$ |
| GRACE | $\underline{76.48 \pm 1.95}$ | $90.22 \pm 0.84$ | $\underline{75.57 \pm 3.01}$ | $87.69 \pm 2.17$ | $\mathbf{66.79 \pm 2.96}$ | $\underline{89.77 \pm 1.59}$ |
| MERIT | $71.70 \pm 2.88$ | $\underline{91.54 \pm 0.75}$ | $72.10 \pm 3.86$ | $\mathbf{94.56 \pm 1.19}$ | $\underline{65.29 \pm 3.23}$ | $\mathbf{91.02 \pm 2.00}$ |
| SUGRL | $\mathbf{84.78 \pm 1.47}$ | $\mathbf{93.01 \pm 0.62}$ | $71.42 \pm 2.68$ | $84.12 \pm 0.75$ | $53.21 \pm 1.80$ | $57.64 \pm 1.79$ |
| TLP with Self-supervised GCL | | | | | | |
| MVGRL | $67.51 \pm 2.21$ | $88.72 \pm 1.04$ | $66.49 \pm 2.75$ | $86.31 \pm 2.09$ | $71.17 \pm 3.04$ | $89.91 \pm 1.45$ |
| GraphCL | $70.26 \pm 2.19$ | $87.32 \pm 1.19$ | $77.26 \pm 3.12$ | $94.13 \pm 1.34$ | $\underline{73.51 \pm 3.18}$ | $\underline{92.38 \pm 1.30}$ |
| BGRL | $64.72 \pm 2.35$ | $90.10 \pm 0.88$ | $68.58 \pm 3.06$ | $89.15 \pm 1.97$ | $60.14 \pm 2.33$ | $79.86 \pm 1.92$ |
| GRACE | $79.38 \pm 1.75$ | $91.68 \pm 0.72$ | $75.23 \pm 2.59$ | $90.48 \pm 1.24$ | $71.21 \pm 2.97$ | $89.68 \pm 1.65$ |
| MERIT | $\underline{85.74 \pm 1.70}$ | $\underline{95.78 \pm 0.61}$ | $\underline{78.14 \pm 3.82}$ | $\underline{95.98 \pm 1.38}$ | $67.67 \pm 2.99$ | $\mathbf{95.42 \pm 1.21}$ |
| SUGRL | $\mathbf{91.63 \pm 1.22}$ | $\mathbf{96.30 \pm 0.51}$ | $\mathbf{85.05 \pm 2.23}$ | $\mathbf{97.15 \pm 0.81}$ | $\mathbf{82.35 \pm 2.21}$ | $92.22 \pm 1.15$ |

**Table 6:** The overall few-shot node classification results of meta-learning methods and TLP with different GCL methods under different settings. Accuracy (↑) and confidence interval (↓) are in %. The best and second best results are **bold** and underlined, respectively.

| Dataset | CoraFull | | ogbn-arxiv | | Coauthor-CS | |
|---|---|---|---|---|---|---|
| Setting | 2-way 1-shot | 2-way 5-shot | 2-way 1-shot | 2-way 5-shot | 2-way 1-shot | 2-way 5-shot |
| Meta-learning | | | | | | |
| MAML | $50.90 \pm 2.30$ | $56.19 \pm 2.37$ | $58.16 \pm 2.35$ | $65.10 \pm 2.56$ | $56.90 \pm 2.41$ | $66.78 \pm 2.35$ |
| ProtoNet | $57.10 \pm 2.47$ | $72.71 \pm 2.55$ | $62.56 \pm 2.86$ | $75.82 \pm 2.79$ | $59.92 \pm 2.70$ | $71.69 \pm 2.51$ |
| Meta-GNN | $75.28 \pm 3.85$ | $84.59 \pm 2.89$ | $62.52 \pm 3.41$ | $70.15 \pm 2.68$ | $85.90 \pm 2.96$ | $90.11 \pm 2.17$ |
| GPN | $74.29 \pm 3.47$ | $85.58 \pm 2.53$ | $64.00 \pm 3.71$ | $76.78 \pm 3.50$ | $84.31 \pm 2.73$ | $90.36 \pm 1.90$ |
| AMM-GNN | $77.29 \pm 3.40$ | $88.66 \pm 2.06$ | $64.68 \pm 3.13$ | $78.42 \pm 2.71$ | $84.38 \pm 2.85$ | **$94.74 \pm 1.20$** |
| G-Meta | **$78.23 \pm 3.41$** | **$89.49 \pm 2.04$** | $63.03 \pm 3.32$ | $76.56 \pm 2.89$ | $84.19 \pm 2.97$ | $91.02 \pm 1.61$ |
| TENT | $77.75 \pm 3.29$ | $88.20 \pm 2.61$ | **$70.30 \pm 2.85$** | **$81.35 \pm 2.77$** | **$87.85 \pm 2.48$** | $91.75 \pm 1.60$ |
| Supervised GCL | | | | | | |
| I-GNN | $68.43 \pm 2.94$ | $78.20 \pm 2.83$ | $65.21 \pm 2.86$ | $77.10 \pm 2.46$ | $65.35 \pm 3.09$ | $76.83 \pm 2.48$ |
| MVGRL | $65.62 \pm 3.11$ | $84.41 \pm 2.35$ | OOM | OOM | $78.08 \pm 3.59$ | $91.78 \pm 1.66$ |
| GraphCL | $60.81 \pm 2.23$ | $81.25 \pm 2.29$ | OOM | OOM | $74.16 \pm 2.88$ | $88.43 \pm 1.73$ |
| GRACE | **$76.78 \pm 3.49$** | **$93.62 \pm 1.32$** | OOM | OOM | $86.22 \pm 2.53$ | $94.11 \pm 1.27$ |
| MERIT | $75.52 \pm 6.53$ | $88.03 \pm 5.11$ | OOM | OOM | $77.52 \pm 7.58$ | **$96.62 \pm 2.12$** |
| SUGRL | $75.98 \pm 2.98$ | $90.02 \pm 1.53$ | **$73.48 \pm 2.55$** | **$81.04 \pm 1.68$** | **$88.45 \pm 1.62$** | $95.10 \pm 0.56$ |
| Self-supervised GCL | | | | | | |
| MVGRL | $78.81 \pm 3.32$ | $91.03 \pm 1.80$ | OOM | OOM | $78.59 \pm 2.92$ | $93.54 \pm 1.40$ |
| GraphCL | $78.49 \pm 3.26$ | $91.32 \pm 2.11$ | OOM | OOM | $78.51 \pm 3.12$ | $91.34 \pm 1.57$ |
| BGRL | $61.08 \pm 2.65$ | $85.03 \pm 2.25$ | $59.91 \pm 2.36$ | $76.75 \pm 0.86$ | $76.85 \pm 3.23$ | $94.69 \pm 1.29$ |
| GRACE | $82.80 \pm 3.13$ | $93.06 \pm 2.17$ | OOM | OOM | $89.46 \pm 2.26$ | $95.53 \pm 1.05$ |
| MERIT | $77.46 \pm 3.14$ | $94.65 \pm 1.31$ | OOM | OOM | $94.31 \pm 1.73$ | $98.35 \pm 0.57$ |
| SUGRL | **$87.98 \pm 2.72$** | **$95.81 \pm 1.69$** | **$82.45 \pm 2.94$** | **$91.68 \pm 1.57$** | **$96.81 \pm 1.31$** | **$98.90 \pm 0.48$** |

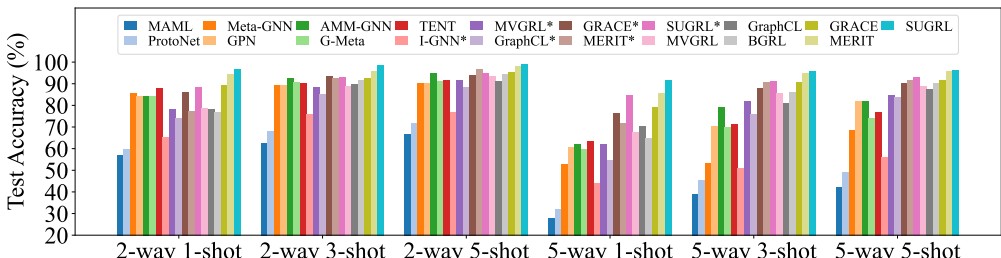

**Figure 9:** $N$-way $K$-shot results on `Coauthor-CS`, meta-learning and TLP. TLP Methods with $*$ are based on supervised GCL methods and I-GNN.

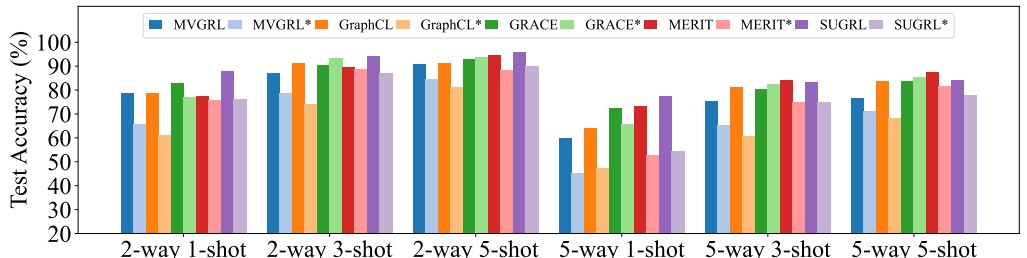

**Figure 10:** $N$-way $K$-shot results on `CoraFull`, TLP with self-supervised and supervised GCL. TLP Methods with $*$ are based on supervised GCL methods.

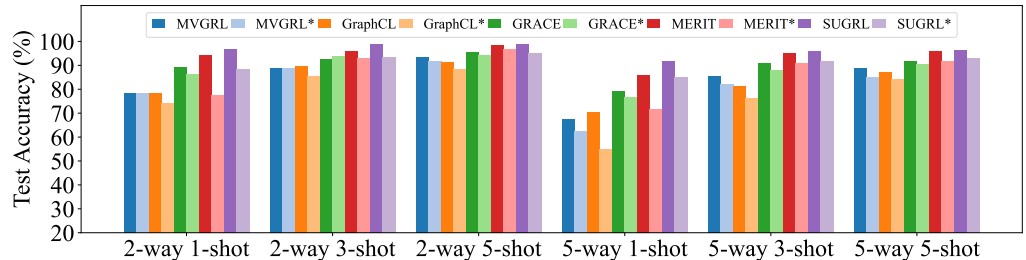

**Figure 11:** $N$-way $K$-shot results on `Coauthor-CS`, TLP with self-supervised and supervised GCL. TLP Methods with $*$ are based on supervised GCL methods.

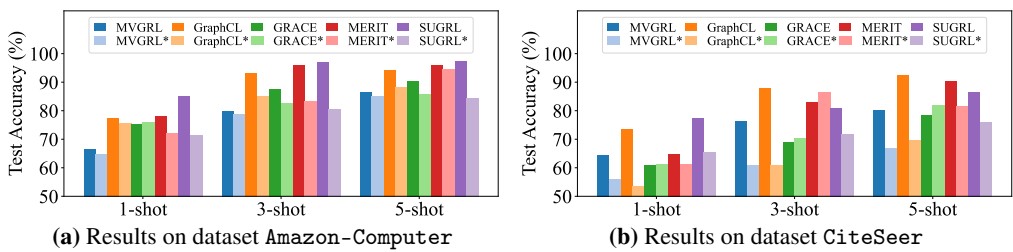

**(a)** Results on dataset `Amazon-Computer`  **(b)** Results on dataset `CiteSeer`

**Figure 12:** 2-way $K$-shot results on `Amazon-Computer` and `CiteSeer`, TLP with self-supervised and supervised GCL. TLP Methods with $*$ are based on supervised GCL methods.

