# OpenReview forum: "Transductive Linear Probing: A Novel Framework for Few-Shot Node Classification"
_logconference.io/LOG/2022/Conference — LoG 2022 Oral_

### Official Review · Reviewer_HuPq · 2022-09-27

**Overall Score:** 8
**Confidence:** 3

**Review:**

This paper motivates and empirically shows the better performance of transferring pre-trained node embeddings (called Transductive Linear Probing) for few-shot node classification, compared to the popular episodic meta-learning method. Extensive experimental evidence and ablation studies have been provided to support the main claim.

---
[+] This paper conducts extensive comparisons of two method classes: (i) (supervised or self-supervised) pre-trained node embeddings and (ii) episodic meta-learning for the problem of few-shot node classifications (FSNC). The main result can have a big impact on the related domain.

[+] The paper is clearly written and easy to follow. The reviewer finds the motivating example is helpful.

[-] It is interesting that TLP with supervised GCL is generally worse than TLP with self-supervised GCL. The authors should explain the counter-intuitive performance in more detail. The results are expected to be better with more information (labels for support classes).

[-] In Figure 2 and Figure 3, with more labeled samples in each support class, we can find that the performance is getting better, which seems to contradict the evidence that "TLP with supervised GCL is generally worse than TLP with self-supervised GCL".

---
After all, the reviewer finds the paper insightful and thorough. This work will be a strong submission if the above confusion can be solved.

---

### Official Review · Reviewer_P5Fd · 2022-10-02

**Overall Score:** 8
**Confidence:** 5

**Review:**

**Summary** -
The authors propose a new solution to the problem of few-shot node classification (FSNC). They deviate from the recent SOTA literature on using meta-learning based approaches which have been shown to work well across multiple modalities. They highlight two particular caveats of meta-learning based approaches - (i) “piecemeal knowledge” that hinders generalizability particularly in graphs and (ii) the existence of sufficient amount of base classes and labelled nodes. The proposed solution named “Transductive Linear Probing (TLP)” operates by learning the representations of nodes in a self-supervised manner. The learned representations are then transferred to the downstream task of learning over the novel classes with only a linear classifier. The procedure amounts to the standard paradigm of pretraining-finetuning which has been empirically verified through a large array of works with theoretically grounded justifications in recent years. Rather than explicitly emphasizing technical novelties in the architecture, the paper has composed a thorough analysis of the fusion of graph contrastive learning with the simplest function, ie, the linear model. The authors perform experiments over several benchmarks and a diverse suite of baselines models and GCL methods.

**Strengths and Weeknesses** -
1. The empirical results are thoroughly justified and the proposed composition of GCL with linear probing beats the previous SOTA approaches by a substantially large margin across the board.
2. The proposed solution is rather simple and can fuse any GCL technique with any type of classifier. I consider it as a strength of the work.
3. The plots provided in figure 5 particularly convince and justify the strength of the proposed framework.
4. It is particularly intriguing that the self-supervised GCL methods beat the supervised GCL methods by a substantial margin, even when N is large. This is an important and useful study for the subsequent works in the literature of FSNC.
5. While the study around the number of classes and labelled examples per class is thorough, I believe there should be more emphasis on studying the variation of the results and the sensitivity against the hyperparameters.
6. Using few-shot TLP as a new metric to evaluate the extrapolation ability of GCL methods is an interesting direction. I support this argument.
7. Although the empirical analysis is extensive, the results clearly verify that a single method isn’t the best across the board. It will be interesting to hear the authors’ viewpoints on how to select the best combination of a GCL technique and the TLP for a given dataset, without comparing all possible GCL techniques. Do the authors have potential solutions to address this?

**Questions and Comments** -
Mismatch between equation and the analysis in section 3.5 . The joint loss in line 259 $L_{JointCon}$ is defined as $\lambda L_{SupCon} + (1 - \lambda) L_{SelfCon}$. However, the explanations in lines 268 and 274 are not inline with this. In line 275, the authors write - “totally supervised GCL (i.e $\lambda = 0$)”, however $\lambda=0$ corresponds to the self-supervised setting based on the definition in line 259.

---

### Official Review · Reviewer_cczC · 2022-10-21

**Overall Score:** 6
**Confidence:** 4

**Review:**

This paper studies few shot node classification in graphs using features obtained with pretrained models. The authors show with experimental evidence that pretrained model embedding, in particular those using graph contrastive approach, can outperform meta-learning approaches.

The paper presents experimental results
-  on the number of novel-classes, number of examples/novel class and its impact in different approaches,
- on different combinations of supervised and self-supervised loss for pretraining,
- evaluating the clustering accuracy of embedding learned by different approaches, and
- visualizing the embedding obtained with different approaches.

Feedback and comments:
- The paper does a good job in terms of experiments by standardizing comparison across methods and datasets.
- However, the technical novelty of the paper is limited in that the work is essentially an evaluation of graph contrastive learning for few shot classification.
- The paper has several observations and ideas from an empirical point of view, since the components of the approach are similar to what has been previously proposed. The authors could improve on the novelty by maybe tying back some of these to the model design.
- The presentation of the paper can be improved. For e.g., [11,12,13] all proposed the pretraining and fine-tuning paradigm - How is the proposed method different? L:73 says this paper presents a comprehensive review on current literature but later L:128 says the review is beyond the scope of the paper.
- The authors claim a transductive approach: Does the contrastive pretraining/supervised training makes use of both base and novel classes? [13] works in a non-transductive setting where the training is only based on the base classes. L:143 claims I-GNN works under a similar non-transductive setting. L:162 then claims TLP is similar to I-GNN and varies only in the pretraining strategy - where does the transductive part come in TLP? Please clarify.
- How is the proposed Transductive Linear probing with supervised contrastive learning different from [11]? Why is [11] not part of the comparison? The  premise, motivation, and observations from [11] are very much relevant to this work - Please position and differentiate your work with respect to [11]. Is the novelty of proposed approach evaluation with self-supervised contrastive learning?
- Is L:243 surprising? I assume this is expected.
- Is scalability the only weakness of TLP method over meta-learning approaches - Please clarify. Can the authors provide observations related to runtime?

---

### Official Review · Reviewer_FMy2 · 2022-10-22

**Overall Score:** 8
**Confidence:** 3

**Review:**

**Summary:**
The paper proposes a comprehensive empirical study that compares meta-learning methods to graph contrastive learning methods in the context of few-shot node classification. In particular, the authors demonstrate and argue for the superiority of representation learning methods in this context.

**Strengths:**
1. The paper tackles the few-shot node classification problem, which is very important and real, for many reasons that the author nicely summarize. Given the new trends in graph representation learning, this analysis provides an updated perspective on how these techniques can be utilized for the purpose of few-shot node classification. This work is reflective and would certainly be of interest to many in the community, it provides a great high-level view of the field.
2. The paper proposes an exhaustive empirical benchmark, which shows a clear advantage for representation learning methods in the graph domain. The analysis is very thorough including multiple methods as well as many datasets of different sizes. The authors provide the code used to run all the experiments, the work done here can be a valuable basis for research in this direction.
3. The authors do a great job at analyzing the results and are very thorough. The writing is very insightful.

**Weaknesses**
1. The study conducted in this paper is clearly tailored to graphs, there is no questioning the contributions there. The claim that this is a new paradigm is, however, not justified. The idea of training on labeled data and then tuning on a few number of samples is not new and has been introduced in previous work across many domains [1]. How do the authors explain the differences? and why does this warrant coining a new term?
2. One key limitation of this work is the lack of multi-graph datasets in the benchmark.
3. [Minor] The use of the term “transductive” is a bit confusing. The datasets that were tested have each a single large graph, so during training the self-supervised method observes the full graph, and these datasets are typically tested in an “inductive” setting, so what makes this “transductive” linear probing? Could this possibly be confusing?

[1] Chen, Ting, et al. "Big self-supervised models are strong semi-supervised learners." Advances in neural information processing systems 33 (2020): 22243-22255.

---

### Meta-Review · Area_Chair_DEQC · 2022-11-18

**Confidence:** 4
**Recommendation:** Accept

**Meta Review:**

This paper proposes a comprehensive empirical study that compares meta-learning methods to graph contrastive learning methods in the context of few-shot node classification.

All reviewers agree that the empirical results are rigorous, strong, and well-hashed out, and that the paper is well-written. One reviewer mentioned a limited technical contribution, but given the strong feedback by all reviewers, I see no reason to go against their recommendations.

---

### Decision · Program_Chairs · 2022-11-22

Accept (Oral)